POINT OF VIEW

# Predictive regulation and human design

**Abstract** Organisms evolving toward greater complexity were selected across aeons to use energy and resources efficiently. Efficiency depended on prediction at every stage: first a clock to predict the planet's statistical regularities; then a brain to predict bodily needs and compute commands that dynamically adjust the flows of energy and nutrients. Predictive regulation (allostasis) frugally matches resources to needs and thus forms a core principle of our design. Humans, reaching a pinnacle of cognitive complexity, eventually produced a device (the steam engine) that converted thermal energy to work and were suddenly awash in resources. Today boundless consumption in many nations challenges all our regulatory mechanisms, causing obesity, diabetes, drug addiction and their sequelae. So far we have sought technical solutions, such as drugs, to treat complex circuits for metabolism, appetites and mood. Here I argue for a different approach which starts by asking: why does our regulatory system, which evolution tuned for small satisfactions, now constantly demand 'more'?
DOI: https://doi.org/10.7554/eLife.36133.001

**PETER STERLING***

***For correspondence:** psterlin@
gmail.com

**Competing interests:** The author declares that no competing interests exist.

## Introduction

*Homo sapiens* is now beset by multiple difficulties. Atmospheric carbon dioxide is rising and destabilizing the climate. Obesity is increasing – although multitudes still live in extreme poverty – accompanied by increases in cases of hypertension and type 2 diabetes. Addiction to opioids and other drugs is also rising, resulting in more deaths from overdose. Medical science continues to advance but, at the same time, the costs of treating an increasingly unhealthy population continue to rise. These problems seem, prima facie, separate. After all, what could possibly connect climate change to obesity, or obesity to drug addiction?

Here I suggest that these difficulties arise from one profound tension: a conflict between how we evolved to live versus how we live now. For 200,000 years we were governed by a close matching of resources to needs. We, like all our progenitors, were shaped by nature's broadest constraint: natural selection favors organisms that gather and use energy efficiently. Everything, from proteins and cells up to organs and the behavior of whole organisms, evolved to be efficient – and to be efficiently regulated by an efficient brain (*Sterling and Laughlin, 2015*).

But then in a flash – the past 250 years – everything changed and many of us now live amidst resources that greatly exceed our biological needs. Although such wealth might seem an unalloyed blessing, it may be also problematic.

What aspect of the design of *H. sapiens* has delivered us to our present state? The short answer, I argue here, is our preeminent computational capacity (that is, our superior intelligence). Once unleashed, it led to a particular event that promptly derailed our species. If we could grasp the enormity of that event and its connection to our present ills, we might see broadly what recovery would require. Here I sketch the growth of the computational capacity *H. sapiens* to show how it has been constrained at every stage by energy and to identify the key adaptations to that constraint.

## Cellular origins of human computational capacity

Human computational capacity began to unfold nearly four billion years ago with tiny cells – prokaryotes – containing DNA whose structure encoded information for assembling proteins that catalyze the chemical reactions that supply

energy and synthesize all the materials needed to reproduce the cell. In essence, the cell is an analogue computer where all information is processed by chemistry (*Bray, 2009*). Analogue is most efficient because the processes are graded and thus matched to avoid waste: just the right concentration of each reactant to produce products that will be just right to serve as new reactants – to make the next products, and so on. Prokaryotes accomplish these functions at an energy cost approaching the lower limit set by physical law (*Noor et al., 2010*).

Cellular chemistry is powered by a small molecule, adenosine triphosphate (ATP), that transfers a fixed packet of energy to another molecule, thereby activating its function. The energy boost provided by ATP is small, about 20 $k_BT$ (where $k_B$ is the Boltzmann constant, and T is temperature) in one millisecond, and modestly exceeds the mean energy of thermal noise. Yet, because a protein molecule is continually energized by thermal noise, the energy boost provided by ATP is enough to tilt the protein from 'almost certainly not active' to 'almost certainly active' (*Astumian, 1997*; *Astumian, 2015*; *Motlagh et al., 2014*). A larger energy boost would be wasteful, so the size of the packet of energy provided by ATP is near optimal. This may explain why prokaryotes adopted ATP as the universal energy donor.

Prokaryotes produced ATP by burning sugars to drive an extraordinary nanoscale turbine (ATP synthase) embedded in the cell membrane. This turbine spins at 9,000 rpm, and each revolution spits out three ATP molecules (*He et al., 2018*). The turbine is 90% efficient, meaning that 90% of the energy that goes into the turbine is 'captured' by ATP (*Kinosita et al., 2000*). In short, the proteins in a cell operate with maximal economy: they are excited by no-cost thermal noise, boosted by energy packets of optimal size that are synthesized with near optimal efficiency.

Yet prokaryotes remained tiny – just a few micrometers in diameter. Their genomes also remained small and did not accumulate information. They had encoded all the genes they could afford energetically, so when new genes were needed to adapt to novel conditions, cells were forced to shed their nonessential genes (*Lane, 2014*). What restrained these early cells, both physically and computationally, was the location of their power plant in the cell membrane. Were a cell to enlarge, its membrane would increase as the square of its diameter, but its volume would increase as the cube. Thus, until a cell could expand its power plant, it could not afford to enlarge (*Lane, 2014*).

Then occurred an event so improbable that it was never repeated (*Lane, 2014*). One bacterium invaded another and took up residence. The host provided nutrients and the guest supplied energy-bearing compounds needed to support its own information system. As long as the host and guest remained independent there was no advantage to the host because the guest used all its energy to reproduce. But gradually the guest transferred most of its genes to the host genome, retaining only key genes needed for substrate oxidation. Now that the guest was a power plant with no household to support, it could multiply without limit and increase the host's energy capacity – by up to 100,000-fold.

The guest, having ceded its passport (most of its genome) to the host cell, became a permanent resident, an obligatory cell organelle: the mitochondrion. Cells powered by mitochondria could now balloon from 3 micrometers in diameter to 300 micrometers, 100-fold greater diameter and 10,000-fold greater volume. These cells – eukaryotes – added 3,000 new gene families along with a more complex and expensive system of gene splicing. Moreover, eukaryotes were under less selective pressure to prune genetic material added by gene duplication or viruses. Consequently, odd sequences accumulated – like bits of wire and old screws in a capacious toolbox. Such fragments were exploited over the next 1.5 billion years by eukaryotes relieved of the prokaryote's energy constraint (*Lane, 2014*).

Unicellular organisms eventually reached hard limits with respect to the capture of resources and information. First, they were constrained from foraging widely by the viscosity of water. For *Paramecium*, rowing through pond water is like *H. sapiens* swimming through molasses (*Purcell, 1977*). Second, despite the invention of subcellular compartments by eukaryotes, there was a limit to how many different proteins could coexist without hindering function (*Zhang et al., 2008*). Third, they grew too large to communicate internally by chemical diffusion (which is fast over short distances but slow over long ones). Fourth, a unicellular organism could send electrical signals across its membrane, but this approach allowed only a single communication

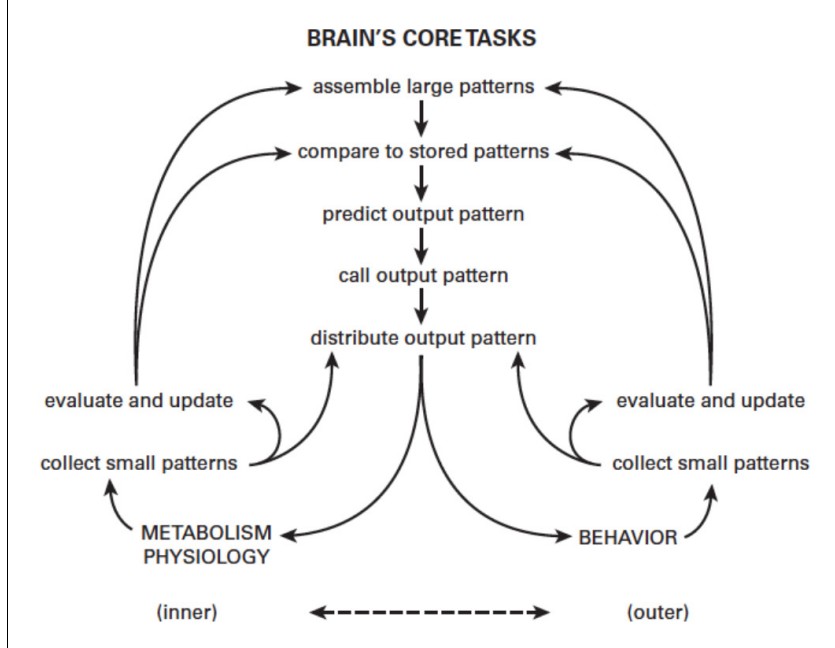

**Figure 1.** Predictive regulation (allostasis). The brain's fundamental challenge is to match the inner needs of metabolism and physiology (bottom left) with the outer needs of behavior. Small input patterns directly drive low-level output mechanisms to produce a rapid response (known as a 'reflex'). Small input patterns are also processed and combined to form larger input patterns that allow informed decisions to be made (for example, if gut is empty, send blood from gut to muscle; otherwise, send blood from kidney to muscle). The brain compares these larger input patterns to stored patterns for historical context (what happened last time?) before deciding on a course of action. The list on the top right shows innate needs served by predictive regulation in the earliest bilaterians, exemplified by *Platynereis*; *H. sapiens* has additional innate needs (bottom right). However, both species use the same 'choosing circuit' (which learns by reinforcement of positive reward-prediction errors).

DOI: https://doi.org/10.7554/eLife.36133.002

channel. These problems were all solved when cells assembled as a multicellular organism.

## Multi-cellularity expanded resources and computational capacity

An animal assembled from many cells could enlarge by orders of magnitude and thus overcome the viscosity of water that had restricted foraging. Multi-cellularity also solved the limit on protein diversity because now cells could specialize their proteins for particular functions such as contraction (muscle cells), metabolic storage (liver cells), and so on. Muscle cells could specialize still further to, for example, contract slowly (gut), rapidly (limbs) or rhythmically (heart). Moreover, as more cells specialized for rapid signaling, an animal could greatly expand its

circuits for processing information. This was essential to manage rising complexity.

By about 500 million years before the present, there arose from the jellyfish line a marine worm termed the *urbilaterian* because it was the progenitor of all subsequent bilaterally symmetrical animals, such as insects and vertebrates (*Arendt et al., 2008*). From this worm, thought to resemble its closest surviving descendent, *Platynereis dumerilii*, we inherited some key building blocks: bilateral symmetry, which was efficient for attaching legs; sensors in the head, which was efficient for detecting what is coming and where we are going; a brain in the head, which shortens the input wires; and a brain extended as a nerve cord, which places motor circuits near their effectors and shortens the output wires. We also received various additional principles of efficient neural design (*Sterling and Laughlin, 2015*).

One key design principle was to regulate in a predictive manner. Internal sensors collect detailed information regarding nutrient levels, osmolarity, sexual state and so on, while external sensors collect detailed information from the environment, such as time, temperature, pH, light, the danger of predators, and opportunities for food, shelter and sex. The brain processes all this information, prioritizes needs, establishes efficient trade-offs, and weighs opportunities against dangers. Then it predicts what should serve best and chooses a behavior, plus all the metabolism and physiology needed to support it (*Figure 1*). Predictive regulation, termed allostasis, minimizes the frequency and size of errors; thus it is intrinsically more efficient than homeostasis, which waits for errors to occur and then corrects them by negative feedback (*Sterling, 2012*).

The earliest key to predictive regulation was a circadian clock. Predicting when to forage, it sets metabolism and physiology to catabolic mode, and predicting when to grow and repair, it resets them to anabolic mode. The central clock governs separate clocks in each tissue so that they can predict locally when to turn on; for example, turn on liver before meals and muscle before exercise (*Gerhart-Hines and Lazar, 2015*). This hierarchy of clocks also exists in the fruit fly, so it too was probably present in our last common ancestor, the urbilaterian worm.

Another principle of efficient neural design was to send signals at the lowest possible rate (that is, the fewest bits per second) because it is cheaper and conserves resources. For example, hormones distribute information wirelessly at

low rates without taking up space. Moreover, the final transfer of information by a hormone (via a G protein) takes about the same amount of energy as is provided by one molecule of ATP (*Sterling and Laughlin, 2015*). To allow signals to be sent at higher rates the early brain added electrical signals, but the energy needed to send a sodium current through a single membrane channel for one millisecond to create an electrical 'spike' or signal takes 2,000 times the energy needed for a hormone signal (*Perge et al., 2012*; *Sterling and Laughlin, 2015*). All this means that the early brain used electrical signals sparingly, and this principle has been preserved during evolution.

Another principle was to learn. By storing experiences an animal can predict what behaviors it should repeat. Thus, when a behavior delivers a result better than predicted, certain neurons reward the 'choosing circuits' by secreting a pulse of dopamine. This encourages the brain to store the memory and repeat the behavior (*Glimcher, 2014*; *Schultz, 2015*). This type of reward learning is mathematically optimal, and since it is present in the fruit fly, it was probably already present in the brain of our last common ancestor the urbilaterian worm. Reward learning emerged early as a core principle of efficient regulation, and *H. sapiens* took it to a whole new level partly because we experience the pulse of dopamine as a pulse of satisfaction – a brief uplift in mood.

## To occupy the world *H. sapiens* required a large, efficient brain

Fast forward to primates. Our nearest living relative, the chimpanzee, forages for visible edibles but lacks the imagination to harvest what it cannot see. Human foragers learned that tubers concealed beneath the ground store carbohydrates that are highly nutritious if properly processed and cooked. Moreover, humans remember where the tubers grow and when. The chimpanzee hunts cooperatively for small animals, but the dominant male immediately appropriates the prize. Human hunters learned that effective cooperation requires fairness, so they manage more restraint and complex schemes needed to capture larger game. The chimpanzee ranges over about 6 km$^2$ of forest, but a human hunter learns to hunt effectively and safely over 12,000 km$^2$ (*Kaplan and Robson, 2002*).

Human foragers collect and process roughly five-fold more calories than a chimpanzee – and do so anywhere on earth. The requirement is a three-fold larger brain that matures gradually. Whereas a chimpanzee has learned all it needs to know for caloric production by age five, a human forager's passage from net consumer to net producer requires 20 years. Moreover, its learning curves for gathering and hunting continue rising until around age 45. By then a hunter will have quadrupled their productivity compared to their entry level. Such prolonged learning indicates that gathering and hunting are challenging careers that require life-long study and practice (*Kaplan and Robson, 2002*; *Gurven et al., 2017*).

The expanded brain continued to follow a number of principles in the interests of efficiency:

- *to specialize,* because two parts are more efficient than one part doing two jobs: for example, visual cortex carves out separate circuits for color, motion, faces, and objects.
- *to only express circuits that are needed,* because this saves space and energy: for example, the visual cortex expands circuits to analyze a tiny, high-resolution patch of the central retina (0.1%) for identifying faces, and it reduces circuits for analyzing the extensive low-resolution regions of peripheral retina. This design shrinks space and energy costs by1000-fold (*Akbas and Eckstein, 2017*).
- *to separate the neural circuits*, because this reduces wire length: for example, visual cortex separates circuits for the two eyes because interlacing them would require each to route its wires around the other (*Chklovskii and Koulakov, 2004*).

Ultimately cerebral cortex of *H. sapiens* subdivided into about 180 different areas (*Glasser et al., 2016*). These include large areas for early sensory processing and motor control, but also various small, narrowly specialized areas, such as six patches for recognizing faces (*Tsao et al., 2008*). Every square millimeter of cortex is occupied with some specific computation. Moreover, the areas retain considerable plasticity. For example, as we learn to read, even in adulthood, a particular area in the left hemisphere shifts from recognizing objects to recognizing written words (*Nakamura et al., 2012*).

Once an individual brain reaches a set volume and cortical expanse, its computational capacity can still increase. First, it specializes the corresponding areas in each hemisphere. For

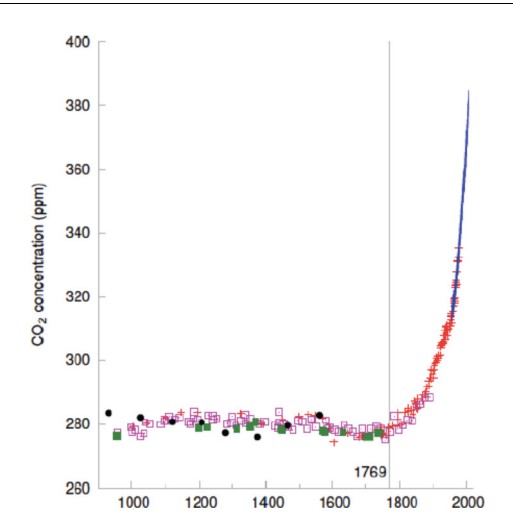

**Figure 2.** The steam engine initiated a sharp rise in atmospheric carbon dioxide. Concentration of carbon dioxide in the atmosphere (y-axis) as a function of year. The steam engine was patented in 1769.
DOI: https://doi.org/10.7554/eLife.36133.003

example, auditory areas on the left emphasize language, and on the right they emphasize music (*Garcea et al., 2017*). Second, it programs cortical areas to mature and regress at different stages of the life cycle, thereby matching expensive cerebral resources to the needs and capacities of each stage. For example, visual and motor tracts mature by our mid-30s and slowly regress, but prefrontal and temporal tracts mature around age 45 and regress much later (*Yeatman et al., 2014*).

Why would *H. sapiens* postpone its final brain development for so long? These late-developing tracts in prefrontal cortex – which support insight, planning, impulse control and choice – ripen in foragers as their economic productivity is peaking. This timing means that a family benefits from a family member during the last few decades of his or her 'three score and ten'. It surprises us to realize that early humans managed to survive so long without modern medicine. So here's the math: each child, on its way to caloric self-sufficiency at age 20, accumulates an enormous energy debt to the family. Partial repayment occurs during the 20s and 30s via calories transferred to the children; but that does not suffice to retire the debt, so repayment continues from ages 40 to 70 via transfers to grandchildren (*Kaplan and Robson, 2002*).

Grandparental foraging allows parents to bear their next child sooner without endangering the existing offspring. If most foragers were to die before repaying their caloric debt, the reproductive capacity of *H. sapiens* would plummet and threaten its extinction. In summary, to occupy the world required a large brain which, in turn, required a long life-span to allow substantial energy transfers from grandparents. Chimpanzees, by the way, repay their smaller debt sooner and are dead at 45 (*Blurton Jones et al., 2002*; *Kaplan and Robson, 2002*; *Kim et al., 2012*; *Gurven et al., 2017*).

## Expanding the community's computational capacity

Once the raw computational capacity of an individual brain reached its limits of space and energy, our species could still expand the computational capacity of the group. Just as the brain specialized the hemispheres for different tasks, it also specialized people, providing each with a different set of circuits and thus a different bundle of innate talents. The learning system encourages the exercise of talents, because that is most rewarding, and soon this generates a community of experts – hunter, healer and so on. A community of experts can easily outcompete a community where everyone is the same, but it can also create profound psychological stresses and interpersonal conflicts.

Therefore the 'community-of-experts' design also includes innate capacities to release tensions and preserve social cohesion. Such behaviors might collectively be termed sacred practices, where 'sacred' means 'reverence for what cannot be expressed in words'. These practices include sex, music, dance and a multitude of ceremonies surrounding birth, puberty, marriage and death. They also included stories, jokes and, eventually, literature. Sacred practices elicit intense emotions such as awe, joy and grief – which somehow bring relief. Accepting the principle 'to use expensive circuits only as needed', the investment of *H. sapiens* in circuits for producing and processing music and art suggest their importance to our design.

Sacred practices emerged early in the evolution of *H. sapiens*. A skilled artist stenciled hands on a cave wall near the southern tip of South America nine millennia ago; a musician played a bone flute in northern Europe 40 millennia ago; sculptors carved abstract objects and ornaments in southern Africa between 70 and 100 millennia ago. Paintings of similar antiquity in a Spanish

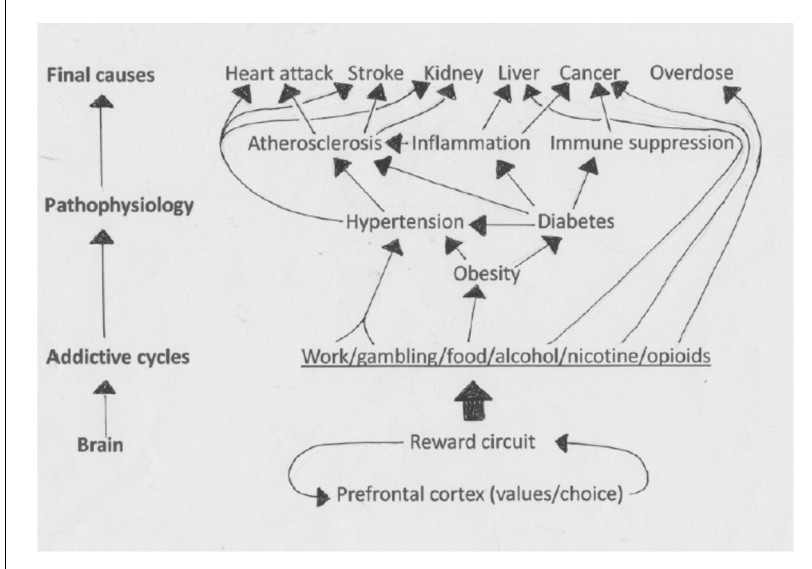

**Figure 3.** The unbounded consumption of rich food drives metabolic circuits awry. The brain drives consumptive behaviors that mobilize many different hormones (not shown) from the brain, gut, liver, pancreas, bone, fat, muscle and other tissue. These regulatory hormones eventually require further elevation: for example, sustained high levels of insulin eventually leads to a need for even higher levels of insulin. The end result can be obesity, diabetes, hypertension and a range of other medical conditions.
DOI: https://doi.org/10.7554/eLife.36133.004

cave are attributed to *H. neanderthal*, suggesting its capacity for sacred practice (*Hoffmann et al., 2018*). Thus, our design – extreme individuality coupled to extreme sociality and relieved by sacred practice – was apparently inherited from our last shared ancestor with *H. neanderthal*.

All these activities could be termed 'culture'. But here I consider them more narrowly – as neurally-generated, species-specific behaviors. The neural circuits that generate and respond to language, music and art required numerous cortical areas to expand at great metabolic cost. That required a richer diet and one that was rendered more digestible by cooking. As these requirements were met, the intestine was reduced. This allowed the brain, which resembles the intestine in metabolic cost, to expand without an overall increase in metabolic rate (*Aiello and Wheeler, 1995*). How language circuits would have paid their way seems obvious – but painting and music? How did the investments in our neural endowment for the arts pay off nutritionally? The answer is that it probably facilitated long-term cooperation between unrelated specialists (*Muthukrishna and Henrich, 2016*).

## What led to our present difficulties?

For roughly two million years members of the genus *Homo* used fire to warm a cave and roast a root. But in 1769 James Watt patented the first efficient machine that used fire to perform mechanical work. This invention coincided with the final stages of 'enclosure' that moved English peasants from their traditional common lands into cities (*Overton, 1996*). The new urban labor pool was immediately set to work at new machines.

Of course, Watt's engineers already had 'slide rules' to calculate his machine's critical specifications; and agricultural productivity had already increased to feed the urban workers. Water-driven mills had already created centers for production, and so on. Yet, the invention that first oxidized carbon to generate power instantly transformed our capacity to exploit resources across the whole planet. Soon the burning of coal enabled steam shovels to dig canals, steam locomotives to cross deserts, steam ships to cross oceans, and steam factories to maintain production, even when rivers froze or went dry. Watt's engine was the most significant development in the generation of power since the mitochondrion.

The steam engine also led to the rise in atmospheric carbon dioxide that drives global warming today (*Figure 2*). Our species' ability to change the climate followed inexorably from the preeminent computational capacity of certain individual brains – like Watt's – and their preeminent capacity to cooperate. But what now drives our consumption of manufactured goods that factories keep pumping out? And what drives our consumption of rich foods and intoxicants? Since biology matched resources to predicted need for billions of years why, in a time of abundance, does human 'need' continue to grow apparently without bound?

The steam engine removed men, women, and children from challenging, integrated lives in the countryside to jobs that yoked them to a machine or sent them down a mine for twelve hours a day or more (*Engels, 1845*). Adam Smith had anticipated the consequence, writing in the very year that his friend's machine went on line: "The man whose life is spent in performing a few simple operations has no occasion to exert his understanding or to exercise his invention in finding out expedients for difficulties which never occur. He naturally loses, therefore, the habit of such exertion and generally

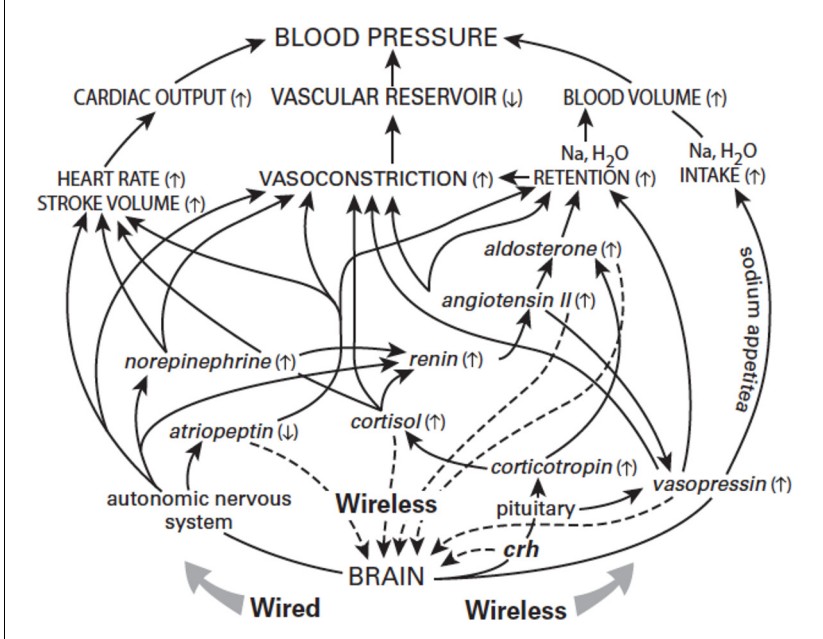

**Figure 4.** The origins of high blood pressure. The brain predicts what blood pressure will be needed and then relies on multiple mechanisms to adjust blood pressure accordingly. These mechanisms, operating on different time scales, are shaped by an interconnected network that employs wired signals (sent along neurons), wireless signals (transmitted by hormones), and motivated behaviors, such as an increased appetite for salt. Again, unbounded individual consumption can lead to high-blood pressure.
DOI: https://doi.org/10.7554/eLife.36133.005

becomes as stupid and ignorant as it is possible for a human creature to become" (*Smith, 1776*).

Many factory and agricultural tasks still fit Adam Smith's description. As do many 'post-industrial' tasks that confine descendants of large-brained foragers within temperature-controlled offices to sit still and stare at screens. Of course many professions – science among them – remain challenging. Yet, it should concern us that simple tasks learned quickly disallow the continuing rewards from prolonged mastery; that time pressure and isolation disallow the rewards from companionate work and community approval; that reduced physical exertion attenuates the rewards from rest; that vicarious sports, music art, and sex lose the rewards from directly exercising wit and skill. Overall, it seems that in 'controlling nature', we shrink the rewards that accompany relief from its small discomforts (*Sterling, 2016*).

What happens when the abundance, comfort and social isolation of modern life shrink the sources of positive surprise required by our ancient circuit to deliver its reinforcing pulse of dopamine? According to computational theories

of reward learning, the system is compelled to seek reinforcement from their intensification (*Sutton and Barto, 1998*). Thus, for many people the main source of positive surprise becomes 'more'. In predicting a Mac, the next surprise must be a Big Mac, and then a Whopper. Notice that when the surprise is more, the next surprise must be even more.

## Predictive regulation in a regime of shrinking positive surprise

Unbounded societal consumption of manufactured goods has been explained by theories that an economy must 'grow or die'. But for a system driven by the need to intensify, 'grow *and* die' is more likely – because unbounded consumption drives diverse, mutually reinforcing pathologies. At the planetary scale, rising carbon dioxide increases atmospheric temperature that melts sea ice. That increases absorption of solar radiation – a cause of further warming that melts permafrost, releasing sequestered greenhouse gases, and so on. The consequent environmental pathologies are innumerable and familiar (*Kolbert, 2015*).

Similarly, unbounded individual consumption of rich food drives metabolic circuits awry. Normally the brain predicts the glucose level needed to fuel, say, a game of tennis and sets the level accordingly by modulating the secretion of insulin and many other hormones. But, when the reward system drives consumption of carbohydrate and fat far beyond metabolic need, the regulatory circuits chronically predict a need to elevate insulin (*Kleinridders et al., 2014*). Insulin receptors in many tissues, including the brain, adapt by reducing their sensitivity to insulin ('insulin resistance'), and this eventually causes cells to need more insulin, which evokes further resistance. The process leads finally to type 2 diabetes, whose complex pattern of endocrine signaling contributes to hypertension and vascular inflammation – thus elevating cardiovascular mortality (*Figure 3*; *American Heart Association, 2017*).

Unbounded consumption of drugs (such as nicotine, alcohol, cocaine, amphetamine and opiates) easily drive the reward circuit into an addictive cycle because they directly release dopamine or prolong its presence at synapses (*Keiflin and Janak, 2015*). Various loops in the reward circuit adapt to these drugs and demand higher doses. Investigators of addiction now recognize that compulsive consumption of rich foods and dopamine-elevating drugs share the

same circuits (*Murray et al., 2014*). And suspicion grows that the same is true of other compulsive behaviors (such as addiction to gambling, porn and shopping). Individuals who are able to exercise their diverse skills, especially those skills for which they have an innate talent, are more likely to obtain sufficient rewards naturally and thus avoid becoming trapped in addictive behaviors.

One doubt raised against the present hypothesis is that, after all, the 'market' has produced wealth undreamed of by earlier generations. It is true that the 'masses' now have access to immense varieties of processed food, processed entertainment, and opportunities for programmed travel. However, these activities are largely passive. Moreover, since they require neither exercise, creativity or courage, they cannot deliver surprises equivalent to the same activities carried out actively. Each one of the ten thousand items in a modern supermarket might potentially deliver a positive reward signal, but these items are all basically predictable – you know exactly what you will find in a particular aisle in a supermarket. Moreover, the ten different brands of, say, olive oil are nearly indistinguishable. Thus the supermarket stimulates boundless consumption precisely because it offers few positive surprises.

## Alternative hypothesis to explain the planetary rise in obesity

Some observers suggest that, when food was more difficult to obtain, we were mostly hungry, so the best survival strategy would have been to gorge whenever possible and store fat via metabolic pathways programmed by 'thrifty genes'. Yet, contemporary foragers (hunter-gatherers) are not mostly hungry. Rather they employ mathematically optimal foraging strategies to satisfy their nutritional needs with relatively few hours per week, leaving considerable free time for sacred practices. Put another way, foragers reliably obtain sufficient food to support their cultural activities (*Kelly, 2014*).

Feeding is regulated meticulously by myriad signals from gut, liver, muscle, fat, bone and brain (*Garfield et al., 2015*). Neural circuits include various 'push-pull' mechanisms that slacken the drive to eat as satiety approaches and restore the drive to forage even as a meal is winding down. Consequently animals in the wild, including *H. sapiens*, tend neither to overeat nor over-forage. They just find other activities. Foraging, remember, carries risk of being foraged.

Where animals do get fat is in a zoo, and that is where we seem to have placed ourselves.

The 'thrifty gene' hypothesis has largely collapsed. Now, except for regions of periodic famine and warfare, essentially all human populations across the planet – whatever their genotype – are rapidly depositing fat (*GBD 2015 Obesity Collaborators et al., 2017*). Moreover, fruit fly larvae fed a high-carbohydrate diet also develop hyperglycemia, insulin resistance, type 2 diabetes and high plasma levels of triglycerides and free fatty acids (*Musselman et al., 2011*). Thus human mechanisms of metabolic regulation were apparently inherited from our shared urbilaterian ancestor, suggesting that all animals are vulnerable to obesity.

When reward diversity is restricted by socioeconomic inequalities, which include poor education and unrewarding 'jobs', eating remains. This may explain why obesity is highest in countries with greatest inequality of income, with the United States literally taking the cake (*Wilkinson and Pickett, 2010*). And within the United States, obesity is highest in the most unequal states, and highest among the least educated (*Ogden, 2010*).

## What level to treat?

The standard medical model views much pathology as biochemical 'dysregulation' – too much of one molecule or too little of another. Consequently therapies focus on drugs to treat biochemical circuits at the lowest levels. For example, the resemblance of binge eating to opioid addiction suggests an antagonist of the μ opioid receptor as a therapeutic agent (*Ziauddeen et al., 2013*). More broadly, newly discovered molecular regulators of carbohydrate and lipid metabolism are often mentioned as possible therapeutic targets for diabetes, heart disease and the panoply of interconnected pathologies shown in *Figure 3*.

Where a molecule is demonstrably missing, such as insulin in type 1 diabetes, the approach has a clear logic and has achieved great triumphs. But where the approach targets a complex circuit that is incompletely known and not obviously broken, the logic is less compelling (*Sterling, 2014*).

Consider, for example, essential hypertension, a major killer that has been treated pharmacologically for half a century. Children entering school are separated from their families and confined in large groups where they must

remain still, silent and attentive to the voice of authority. Children gifted with those abilities are well rewarded with praise and gold stars. But for children gifted with a compulsion to move and a strong inner voice, school can be stressful, and soon blood pressures start to rise. By age eighteen nearly 50% of US boys have systolic pressures of 130 mm Hg or higher (*Falkner, 2005*). The brain, regularly predicting a need for higher pressure, drives all components of the circuit (*Figure 4*). Gradually each component learns its role in sustaining elevated pressure (like skeletal muscles learning to anticipate exercise). This circuit is not 'broken', nor are the components behaving 'inappropriately'; they are simply responding to the brain's predictions. Eventually, however, the circuit enters a pathological cycle: as arteries adapt to higher pressure by thickening and stiffening, they need more pressure to achieve the required output – but higher pressure further narrows and stiffens the arteries. Finally the system loses the ability to resume normal pressure as it should, for example, during sleep. Chronically high pressure causes inflammation, which encourages atherosclerosis and eventually heart attack or stroke.

In treating hypertension medicine commonly targets the lowest level mechanisms (*Figure 4*). Since one cause of high blood pressure is excessive fluid for the vascular reservoir, hypertension is often treated with a diuretic to shrink blood volume. But then the brain, predicting a need for high pressure, compensates by shrinking the reservoir. To prevent that, a calcium antagonist is added to relax vascular smooth muscle. Still the brain insists that pressure should be high, and again it compensates by increasing cardiac output. To prevent that, a beta blocker is added, thus antagonizing the last pathway capable of raising the pressure. Unfortunately, this also renders the patient unable to exercise – a core need for every aspect of physical, emotional, and cognitive health.

Similar scenarios can be drawn for other epidemic problems, such as obesity, type 2 diabetes and drug addictions. All of our control systems are designed with multiple compensatory loops because that allows for efficient trade-offs. This leads me to doubt that any 'magic pill' will heal the global epidemics whose simple cause is unbounded consumption and whose deeper cause is loss of reward diversity. A more promising strategy would be to re-expand the opportunities for small satisfactions and thus rescue the reward system from its pathological regime. Individuals satisfied by their

work and active extracurricular activities are less likely to rely on food and other substances to quell their restless searching.

To reach this point would require a substantial reorientation of attitudes and politics. However, since the result would reduce consumption of everything from fossil fuels to medical care for chronic disease, economic resources should present no obstacle.

## Conclusions

Just as single cells are designed to operate just above thermal noise, single humans are designed to operate just above the roiling currents of their myriad needs. This means that we are constantly reaching out and struggling to stay above the waves. However, to me at least, the core of the design of *H. sapiens* – extreme individuality coupled to extreme sociality – suggests some principles for healing.

First, recognize individual clustering of abilities and deficits as fundamental to our design. A deficit is not *prima facie* a 'disorder'; it may simply be a gap to be filled by a neighbor, whose own gap may in turn be filled by someone else. We need to develop constructive niches for individuals with different clusters. To start, we should cease confining all children into the same classroom and treating those who tolerate it poorly with amphetamines. Instead, we should identify the special gifts conferred on each child and support their early practice. The resources needed to accomplish this are now being squandered, as later down the line they are spent to incarcerate aimless, angry, drug-addicted young adults.

Second, recognize that pharmacology cannot be the primary route to health. The brain has been regulating the body since *Platynereis*, and the systems are far too complicated to be managed primarily with pills.

Third, re-diversify activities that can offer unpredicted rewards. Much public discourse concerns the need for more jobs. But the inexorable trend toward automation will be reducing the number and quality of jobs. Masses of humans will need something interesting to do besides consuming stuff and traveling from place to place snapping 'selfies'. We will need somehow to reverse the trend that Adam Smith identified in 1776 toward losing our skills and alertness from oversimplifying work.

Fourth, renew and re-diversify the sacred practices upon which *H. sapiens* depends as a species to relieve the tensions caused by our

innate strangeness. This included music, art, dance, literature and monumental constructions that engaged large numbers of people over decades. Of course, we are already rich in the products of sacred practice – we can enjoy a concert, play, museum, or novel – for an hour or so. But it is the artists who practice their sacred skills over years and decades who benefit daily from the small, unpredicted rewards of improving a skill. Vicarious activity cannot substitute for individual engagement.

## Acknowledgments

I thank the following individuals for their encouragement and comments on various drafts of this essay: Dean Astumian, Sally Zigmond, Simon Laughlin, Michael Mullin, Alan Pearlman, Robert Collins, Price Peterson, Irving Seidman, Wolfram Schultz, Gino Segre, Dost Ongur, Scott Poethig, Jonathan Demb, Olivia Demb, Andy Barto, Ann Stuart, Guy Lanza, Jose Tapia Granados, Nica Borgese, Stan Schein, Richard Masland, Bart Borghuis, Kai Kaila, Marla Feller and Adrianne Fairhall. I also thank Iris Broudy for preparing the manuscript.

**Peter Sterling** is in the Department of Neuroscience, Perelman School of Medicine, University of Pennsylvania, Philadelphia, United States. He also farms citrus and coffee in the western highlands of Panama

psterlin@gmail.com

http://orcid.org/0000-0001-5010-8907

*Competing interests:* The author declares that no competing interests exist.

## Decision letter and Author response

Decision letter https://doi.org/10.7554/eLife.36133.008
Author response https://doi.org/10.7554/eLife.36133.009

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
