## [Decision Letter]

Thank you for submitting your manuscript "Neurobiology of Homo sapiens: what went wrong?" to *eLife* for consideration as a Feature Article. Your manuscript has been reviewed by three peer reviewers, and the evaluation has been overseen Peter Rodgers as the *eLife* Features Editor.

The full reports from the reviewers are below. The following individuals involved in review of your manuscript have agreed to reveal their identity: Jeremy Gunawardena (Reviewer #2) and Bruce McEwen (Reviewer #3).

While the reviewers found the manuscript to be interesting and provocative, they also found it to be "speculative and superficial" in places. It is vital, therefore, that you fully address all the concerns of the reviewers in a revised version of the manuscript. In particular, you need to address the following points:

– Referee 1. Point 1 (which argues against the conclusion that "modern life shrinks reward diversity.")

– Referee 1. Point 2 (which argues against the conclusion that something has "gone wrong"/that we face a "crisis"). Referee 2 makes a similar point in paragraph 7 of his report.

– Referee 2. The comments in paragraph 4 (about extreme teleological language).

– Referee 2. The comments in paragraph 8 (about language and culture).

– Referee 2. The comments in paragraph 9.

– Referee 3. The comments in paragraph 2 (about the role of the human prefrontal cortex)

*Reviewer #1:*

This article discusses the origin of various "crises" impacting the human race (climate change, obesity) and argues that they all originate from the drive for positive reward prediction error, which in a society of abundance results in excessive consumption. The article is interesting and provocative in a popular science way, and presents many intriguing connections between evolution, neuroscience, history, and behavior that will stimulate thought among readers. But it is also extremely speculative and superficial in its treatment of these topics.

Major points

1) This is a strange article in that it is extremely broad, speculating about everything from the evolution of life on earth to the modern opioid epidemic. I am not sure that something like this belongs in a scientific journal that publishes original research such as *eLife*, or at least I am not sure by what criteria it should be reviewed. The article skips through so many different areas of biology at such a superficial level that it is unclear where to begin with criticism. For example, a central conclusion is that "modern life shrinks reward diversity." Is this true? I could make a counterargument that life as a hunter-gatherer was likely repetitive, and that my modern existence as a scientist living in San Francisco provides a much greater diversity of interesting and unexpected stimuli. But in any case the author provides no evidence for his position beyond examples and anecdotes.

2) The article starts from the premise that something has "gone wrong" with the human race and that we face a "crisis". But by virtually every measure the human race is better off now than it has been at any point in history – longer lifespans, better health, greater freedom and security, etc. The "crisis" that the author describes consists of a few cherry-picked examples that support his thesis. It is reasonable to suggest that obesity or drug addiction may be caused by dysregulated reward systems, but the author instead wants to propose an overarching theory for life on earth.

*Reviewer #2*:

1) This is a fascinating and ambitious paper, which contains many interesting ideas that should be of broad interest. However, it also contains suggestions that are poorly worked out, which get in the way of appreciating the former.

2) The author attempts something that is increasingly rare, which is to integrate our scientific understanding across the scales from the molecular to the physiological to the social. I can think of few other scientists with the chutzpah to attempt this, with the exception perhaps of Jean-Pierre Changeux ("Climbing brain levels of organisation from genes to consciousness", TICS 21:168-181 2017). This kind of integration is sorely needed, because the rest of us are too scared to attempt it, and I vigorously applaud the author for trying.

3) The most interesting and most important ideas centre around "allostasis", or the re-evaluation of the classical homeostasis of Claude Bernard and Walter Cannon to incorporate the central nervous system, prediction, motivation and feed-forward regulation (Figures 3 and 11). This draws on the author's own seminal contributions (Sterling, 2012), as well as the powerful synthesis of neural "design principles" in the author's collaboration with Simon Laughlin (Principles of Neural Design, MIT Press 2015). These ideas suggest that the organism is not just responding to its environment, as in classical homeostasis, but actively, if largely unconsciously, exploring and internalising it. The organism acquires thereby an unconscious autonomy over aspects of its functioning that has previously been overlooked (Figures 10 and 12). This is an extremely provocative viewpoint and I quite understand why the author feels it ought to have profound implications for how we think about the human organism in the context of upbringing, education, illness and participation in society.

4) Allostasis is described within an account of how increasing information-processing complexity has emerged from the first primordial cells to us (first 3 sections after the Introduction). This would be engaging and entertaining were it not for its extreme teleological language. For example, "Cells began to accumulate odd sequences that could serve some future challenge", "Moreover, an animal could specialize many cells for rapid signaling and thus greatly expand its channels for processing information". What?! Evolution could look ahead, could it, to see what it would need in the future? It will be anticipating the emergence of humans next. Who needs intelligent design when a biologist writes like this? Of course, the author knows better, which only makes it worse. Evolutionary biologists must despair of the rest of us.

5) The clinical context is where allostasis should have the most traction, by providing an alternative to traditional pharmacology's "magic bullet", as trenchantly described in "Treating the lowest levels". Working out a strategy for reaching the "higher levels" and convincing clinicians of its value would be an immense contribution.

6) However, the paper does not take this route. It seeks instead to stretch much further and here the ice starts to crack, if not disappear altogether. According to the author, the human species "went wrong" by discovering capitalism (Figure 8), which "shrank reward diversity" (Figure 9), so that our allostatic biology led inexorably to global warming and a pandemic of ignorance, stupidity, Big Macs, obesity, diabetes, hypertension and addiction. Furthermore, we are all so different that mental disease should be placed on a continuum, rather than treated categorically ("Understanding human differences"). I do not find these assertions compelling for many reasons.

7) First, each of them is worthy of a paper, if not a book, in its own right, which, to be credible, would have to take on board a formidable body of existing literature. In the case of what "went wrong", if, in fact, anything did, the discussion would have to go back at least as far as Rousseau's "noble savage". As they stand, these assertions come across as the author's personal view of the World.

8) Second, there is a long, and mostly unhappy, history of biologists trying to contribute to the human sciences. The author is attempting to do this at a very ambitious level but there is a striking failure to mention the key concepts which differentiate us from other animals and which form the intellectual currency of the humanities – language and culture. Allostatic biology is shared with many other animals. Language and culture are not. The former appears once in the paper, as an example of brain specialisation, and the latter appears not at all. How can they not be part of the explanation? Sociobiology at least attempted to say something about culture, with the "gene" as the unit of explanation, however unwelcome that was to most social scientists, but the author ignores culture, as if to say that allostatic "design", in place of "gene", is now sufficient to explain human behaviours and societies. There is an enormous gaping chasm here, to put it mildly.

9) Finally, these assertions lead to suggestions of what we should do to "resolve the tension between our biology and how we now contrive to live", which are, as listed in the "Conclusions", to "develop constructive niches for individuals", "reverse the trend toward becoming ever more stupid and ignorant", "identify each child's special gifts", "renew sacred practices" and be accommodating of sociopaths. I am afraid these strike me as platitudes.

10) In summary, there is an important and interesting paper here but it needs sharper focus and ruthless editing to extricate it from the author's personal beliefs. The updated view of the organism that the author is putting forward is worthy of serious scientific attention. If it becomes broadly influential, as it should, then perhaps in time it may help lead to the changes in society that the author wants to see. If so, it will be others who accomplish such change, not the author. He does himself, and us, a disservice by seeking a final cause, rather than allowing evolution to take its course.

*Reviewer #3*:

1) Thoughtful, informative of the science, and enjoyable to read. Clearly the author is very passionate about this topic and rightly so!

2) One possible addition, to what is implied but not stated, is to add a discussion of the variable role of the human prefrontal cortex in self-regulation of impulses, mood and other behaviors as well as proactive planning. The PFC develops after birth in formative years of childhood, adolescence and into young adulthood, and it develops more quickly on the average in females. (Hence, why for young males, auto insurance rates are higher than for females!) The variability of results of the famous "marshmallow test" on children indicates how variable among individuals is "self-regulation"; tests on children turn out to be predictive of self-regulations 40 years later! Early life adversity redirects the PFC development and is likely to be a contributor to the variability in impulsiveness, quarrelsomeness, proactive planning (e.g. for the future of our planet) as well as the pursuit of consumption and hedonic rewards.

3) Clearly education is an important factor. Our prolonged developmental period as humans allows the fortunate to develop the perspective needed to do what the author advocates. So many people are not so lucky!

[Editors' note: further revisions were requested prior to acceptance, as described below.]

Thank you for submitting the revised version of your article "Predictive regulation and human design" to *eLife*. I have received comments on the revised version of article from two of the reviewers who reviewed the original version, and discussed the revised version and these comments with another editor.

We agreed that you need to address some of the points made by Jeremy Gunawardena, and that we will accept the article for publication if you address these points plus a number of editorial points.

Re the comments from Jeremy Gunawardena: Please address the following points:

– point 3

– points 4 and 5 (which are related)

– points 6 and 7 (which are related)

– point 8. (Addressing some of the editorial comments should help address this point)

1) The author has made substantial changes to the original draft but not really in the way I was hoping for. Some of the interesting scientific details have been lost (previous Figures 5, 6, 10, for instance) but the personal opinions largely remain. However, the changes have made the logic of the argument clearer and I will focus on that.

2) To summarise, the author's claim is that the dopamine-based, reinforcement learning system which we share with other animals entered a regime of "positive feedback" in which "consumption grows explosively" because "modern life shrinks reward diversity", and this led to our "present difficulties" of "climate change, obesity, drug addiction".

3) Like reviewer 1, I am not convinced that modern life has lower reward diversity than that of an agricultural labourer in 1700 CE or a stone carrier undertaking the "monumental construction" of the pyramid of Cheops in ~2500 BCE. I expect it is quite the reverse. The author does not tell us how reward diversity can be measured and the issue tested, so this remains a matter of opinion. That is part of the problem, especially when personal opinion is not clearly stated as such (see point 8).

4) Even if we give the author the benefit of the doubt over point 3, I do not understand the claim that reinforcement learning systems lead to positive feedback and explosive growth. As the author points out, these learning algorithms have properties of optimality within a specified mathematical context (Niv, J Math Psych, 53:139-54 2009). Positive feedback and explosive growth are pathologies, which would destroy optimality. So, how do they arise?

5) The author suggests that they do so through habituation: "all neurons and circuits adapt. As a result repeats and becomes predictable, its targets at the molecular and circuit levels reduce their sensitivity". That is well and good but habituation leads to levelling off, not positive feedback, and levelling off is what a reinforcement learning system would do if it kept receiving the same reward. No instability arises. The assertion that reinforcement learning systems become pathological is not justified.

6) It is an enormous leap to implicate dopamine reward pathology as the cause of our present crises (I have no problem calling them that). I previously pointed out that culture is the more conventional culprit. I did not mean by culture, "art, music, etc.", but rather the societal infrastructure – language, laws, Government, economy, industry, media, institutions – which has allowed us to dominate the planet. The author has not convinced me that biology is a better explanation for the crises than culture, understood in this sense.

7) To clarify the difficulty, let us consider drug addiction because this is an area in which dopamine-based reward systems may play an important role in individual pathology (Berridge, Psychopharmacology, 191:391-431 2007). But is the reward system the cause of the addiction crisis? During the Opium Wars of the 19th century, the British East India Company used gunboat diplomacy to force the Chinese Qing dynasty to license opium sales in China, resulting in an addiction crisis that affected a large part (some reports suggest 25%) of the population. Was the crisis due to an "explosive" dopamine system in the Chinese or to the British creating the world's first drug cartel? I think the latter is "the cause" and I suspect most commentators would do the same. This is not to say that the biology is unimportant but it seems more plausible that its role is in deciding who becomes addicted, and who escapes, rather than being the root of the crisis, which lies in culture and society (here, British imperialist trade policy).

8) As I said previously, the author has put forward some fascinating and important scientific ideas that deserve wider attention but I fear these are getting lost in the speculation and personal opinion. This is a perspective, of course, and speculation and personal opinions can sometimes be hugely beneficial in stimulating fresh thinking. I have no problem with that but speculation needs to be clearly seen as such and not dressed up as if it were science. The author states in his rebuttal letter that he has recast his views as a hypothesis and seems to feel that his personal opinions have been adequately segregated. I do not agree. I think these are still very much present in the issues discussed above. The main claim is more clearly a "hypothesis" in the rebuttal than in the revision. In the latter, "hypothesis" is used directly only to refer to "alternatives" and only once used, indirectly, to refer to the "present hypothesis". The speculative nature of what is being claimed and the extent to which it reflects personal opinion is not adequately stated in the new draft.

---

## [Author Response]

While the reviewers found the manuscript to be interesting and provocative, they also found it to be "speculative and superficial" in places. It is vital, therefore, that you fully address all the concerns of the reviewers in a revised version of the manuscript. In particular, you need to address the following points:– Referee 1. Point 1 (which argues against the conclusion that "modern life shrinks reward diversity.")– Referee 1. Point 2 (which argues against the conclusion that something has "gone wrong"/that we face a "crisis"). Referee 2 makes a similar point in paragraph 7 of his report.– Referee 2. The comments in paragraph 4 (about extreme teleological language).– Referee 2. The comments in paragraph 8 (about language and culture).– Referee 2. The comments in paragraph 9.– Referee 3. The comments in paragraph 2 (about the role of the human prefrontal cortex)

I heartily thank all three reviewers for their thoughtful comments and vigorous wake-up call. Trying to read the essay through their eyes, I find many instances where rhetorical intensity replaced solid explanation and have revised extensively to fix that. I have also followed reviewer 2’s request for “sharper focus and ruthless editing” – as detailed below. The text plus legends is shorter by about 1000 words.

1) Reviewers 1 and 2 objected to the term “crisis” and to the suggestion that something “has gone wrong”. Accordingly, the Introduction now describes the rise of atmospheric CO_2_, obesity, and drug addictions as “difficulties” caused by excessive consumption.

The Abstract, Introduction, and final section of the essay mentioned “mass migrations” as one of the “crises”. Since migrations are not directly caused by excessive consumption (the main focus), that material is now deleted. Also deleted is the material on human differences and mental disorder.

2) The essay now traces the evolution of computational capacity constrained by energy efficiency (the topic of my book with Laughlin) to what I identify as a critical event. It then hypothesizes a connection between this event and subsequent “difficulties”. Reviewers 1 and 2 may remain unconvinced, but at least the hypothesis now rests more clearly on evidence from the scientific literature and not my personal opinions.

3) The section, now headed “Multi-cellularity expanded resources and computational capacity”, contained the sentences to which reviewer 2 objected as “extreme teleology”, and I have re-phrased them to meet his concern. I also reviewed the rest of the text with this concern in mind, but I am clearly less sensitive to this issue than reviewer 2, so if he finds additional objectionable phrasings, I’m certainly willing to accommodate.

This section describes several principles of efficient design adopted by early brains, including *predictive regulation* (allostasis), where a clock governs a primitive neuroendocrine system and hypothalamus. One principle, *learn,* is introduced as key to predictive regulation along with its efficient mechanism, dopamine reward of better-than-predicted results. This mechanism appeared 500 million years ago with worms, a point that becomes relevant in the essay’s last section.

4) The next section retitled, “To occupy the world H. sapiens required a large, efficient brain”, is reorganized to better explain the degree of computational capacity that humans required to inhabit the entire natural world: a brain 3-fold larger than that of our nearest living relative, the chimpanzee. To grow such a large brain we assume a deep caloric debt that requires a lifetime to repay (70 years). Foraging and hunting skills require such extended learning that productivity doesn't peak until age 45, when foragers are grandparents. Our long life span and economic contributions as grandparents (ages 45-70) are essential to support a birth rate high enough to prevent extinction. This section relies on current anthropology – detailed economic measurements and analysis of modern foragers on three continents with supporting citations. This material represents the current consensus in this field -- not my personal conjecture.

These anthropological data help make sense of the recent MRI finding that tracts for prefrontal cortex that support judgment, choice, planning, and impulse control -- all those responsible behaviors – finally mature during our middle to late 40s and are the last to regress in old age. It seems plausible that foragers would live long enough to benefit from their final brain maturation. The evidence cited does not romanticize the “noble savage”, but neither does it support Hobbes’ description, ‘nasty, brutish, and short”. Reviewer 3 requested additional comment on this prefrontal expansion, and I have added some as part of the section on late-maturing tracts.

5) The next section is also retitled for sharper focus: Expanding the community’s computational capacity. In four paragraphs it notes that specializing individual brains extends our species computational capacity – but entails additional mechanisms for cooperation and relief of tensions. This is where “culture” enters. I address it, as reviewer 2 requested, noting that neural circuits for culture belong to the overall problem of efficient computing – neural circuits for music and art must pay their way. I suggest what seems fairly obvious, that they do so by facilitating long-term cooperation between unrelated specialists. There are undoubtedly many other ways to say this, but mine is at least economical – in the spirit of reviewer 2’s request for ruthless editing.

Various investigators, Boyd and Richerson, for example, discuss co-evolution of culture and genes, but I haven’t found specific reference to the role of culture in enhancing computational capacity by facilitating brain specialization. If reviewer 2 knows a good reference here, that would be great.

6) The next section, “What led to our present difficulties?”, presents the core hypothesis. Since that will be controversial, at least the historical facts should be well accepted – no cherry picking allowed – and for that I draw on a recent authoritative history (Overton). The key effect of factory work was captured by the succinct observation of a contemporary observer, Adam Smith, the father of modern economics. His description contrasts sharply to the life of foragers presented in the preceding section. So far, no personal opinion.

Then comes the essay’s core hypothesis, that modern life shrinks the diversity of better-than-predicted rewards and thus drives the reward learning circuit into positive feedback. The problem is presented more carefully than in the earlier version and hopefully the reasoning is clearer. Even if reviewers 1 and 2 are not ultimately convinced, I believe that the idea is broad and original enough that it deserves consideration – especially since it potentially explains a host of health issues. Thanks to reviewers 1 and 2, the idea now emerges as a hypothesis rather than as sheer opinion larded with a host of other issues.

The next section, **“**Predictive regulation under positive feedback”, explicitly connects the “difficulties” of CO_2_, obesity, and drug addiction as systems gone haywire due to positive feedback. Same facts as before, but organized better to address the broad hypothesis.

Finally, I moved the material on hypertension to the end, where it is useful in explaining problems of treating the lowest regulatory levels with drugs.

Specific comments.

i) Reviewer 1 suggests that a forager’s life was repetitive and that his own “modern existence as a scientist living in San Francisco provides a much greater diversity of interesting and unexpected stimuli”. Reviewer 1 objects that I provide no evidence beyond anecdotes.

The essay now offers quantitative evidence of the long-term challenge of foraging and hunting – activities complex enough to require learning into middle age. A hunter, for example, covering a territory of 12,000 km^2^ seems unlikely to repeat very much – except possibly upon reaching the familiar track to home -- and that would be rewarding.

Of course, rewarding stimuli are numerous for a scientist in San Francisco – and that is exactly my point. Research scientists are indeed rich in diverse activities – probably competing well with a forager in this regard. And correspondingly, our rates of obesity and drug addiction are low. But, if reviewer 1 traded his profession for the job of a supermarket checker, his opinion might change. The essay notes that obesity is most prevalent among the least educated who, if they are employed, have the least rewarding jobs

ii) Reviewer 1 says, “it is reasonable to suggest that obesity or drug addiction may be caused by dysregulated reward systems, but the author instead wants to propose an overarching theory for life on earth*”.*

If we agree that obesity and drug addiction are caused by a disturbed reward system, we can agree further on the importance of asking what causes that “dysregulation”. The essay presents my hypothetical answer.

The essay is not meant to present “an overarching theory for life on earth”. Rather, it simply tries to explain why consumption is rising in the midst of plenty, where normally one might expect satiety. Since the reward system is a core principle of all animal behavior from worms onward, and since it influences every choice under control of the prefrontal cortex, naturally its dysregulation in humans would have broad effects.

iii) Reviewer 2 [point 8] criticizes the failure to consider genes and culture as contributions to the difficulties of concern here. I do discuss the ‘thrifty gene’ hypothesis and its inadequacies. Beyond that I omit these vast topics – because, unlike Sociobiology, the essay does not attempt to unify all behavior, genetics, and culture. To the contrary, my point is more focused: essentially *all* cultures and *all* genetic stocks are struggling with the *same* difficulty––unbounded consumption––so I don't see where culture would apply.

iv) Reviewer 2 criticizes the failure to distinguish our species from animals. I cannot see a reason to make this distinction because the problem lies, according to my hypothesis, with a mechanism that we *share* with animals: the reward system. Rats and fruit flies become obese and drug-addicted just like people; for example, a new paper in Current Biology shows that a male fruit fly, restricted in reward diversity by optogenetic inhibition of its ejaculation, drinks more ethanol. All animals are wired for multiple small rewards, and when the possibilities shrink, they all seek ways to get their hits of dopamine.

I cite no evidence that culture and education protect against obesity and addiction because they don’t – except as they promote opportunities for reward diversity, which I do cite.

v) Reviewer 1 argues that we are better off today than at any time in human history. Maybe so for research scientists in San Francisco, but there are now 8 billion members of the “human race”, and a great proportion are living miserably – even in our great cities. However, I prefer not to enter this debate, which has been treated exhaustively by Steven Pinker (among others), so it does not appear in my essay.

vi) Reviewer 2 objects to the conclusions as platitudes. I claim to the contrary that these social policy suggestions emerge from basic principles of human design set forth in the essay. For one example, the principle: e*nhance group computational capacity by specializing individual brains,* cries out for individualized education. Yet public education is anything but that. Furthermore, the low tolerance of many children for a one-size-fits-all classroom evokes behavior diagnosed as ADHD that is widely treated with amphetamines, which of course are also widely consumed as drugs of abuse. A reform that re-individualized education would require a substantial social investment – so this principled reversal of current social policy goes well beyond a platitude.

More generally, the computational richness accumulated in neural circuits over 500 million years leads me to doubt that the “disorders of consumption” will be cured by new drugs that tweak one or another protein receptor. Since that is the only approach expressed these days in leading journals, a principled critique seems overdue. My critique generally provokes outrage; whereas platitudes provoke a yawn.

[Editors' note: further revisions were requested prior to acceptance, as described below.]

We agreed that you need to address some of the points made by Jeremy Gunawardena, and that we will accept the article for publication if you address these points plus a number of editorial points.Re the comments from Jeremy Gunawardena: Please address the following points:– point 3– points 4 and 5 (which are related)– points 6 and 7 (which are related)– point 8. (Addressing some of the editorial comments should help address this point)

The attached manuscript now:

i) includes your revisions to the Abstract – except for “treating neural circuits […]” because many of the treatments are somatic as well. My point is that the circuits are complex, so I added that word. Also accepted are your revisions to the Introduction.

ii) attends to all the points that you inserted with bold type. These changes also address the reviewer’s concerns (4 and 5) about the terms “positive feedback” and adaptation, which I have deleted. That section now simply describes what is observed, a “pathological cycle”.

iii) omits the figure regarding reward diversity and adds explanation to figure legends where requested.

iv) adds the carbon dioxide curve with a simple legend and attribution.

v) accepts the reviewer’s request (point 8) to better identify my opinions; have done so at all the points that you kindly suggested.

That leaves the reviewer’s three related points. His point 6 states “It is an enormous leap to implicate dopamine reward pathology as the cause of our present crises.” But that is not my claim and does not accurately represent my argument. I doubt that the neural circuits are pathological. They simply respond to the situation, and when that leads to unfortunate behaviors, it is not their fault.

My concern is unbounded consumption. That, undeniably, is what causes the rise in CO_2_; it is what causes obesity (and its sequelae); and it perfectly characterizes the cycle of addiction to various intoxicants. So my question is, what causes unbounded consumption?

Researchers in the field of reward learning agree that the various intoxicants drive dopamine release or increase its dwell time at synapses. They also agree that it is for this effect on dopamine that the drugs are consumed without bound. Many researchers also accept that rich foods are consumed to caloric excess for the same reason––to drive the dopamine system. In fact, several studies propose to treat obesity with opioid antagonists. So the connections between obesity, drugs of addiction, and the dopamine reward system are clear and uncontroversial.

It is also widely recognized that unbounded participation in other activities, such as work, gambling, sex, and so on, also release dopamine and can drive an addictive cycle. Hence the expression “workaholic” and the 12-step programs and support groups for diverse unbounded behaviors.

Authorities on reinforcement learning agree that learning by machines and animals, starts with a constitutive seeking of reward. A computation predicts a behavior, and when the result is better than predicted (positive reward-prediction error), the machine gets a “reward” signal, and the animal gets a pulse of dopamine. It is also agreed that, when sources of positive reward-prediction error shrink, the machine and the animal will seek a familiar reward for which the only positive prediction error is greater intensity; i.e., more. This is in the math and in its recognized biological instantiation.

The above generally accepted points lead me to hypothesize that unbounded consumption in modern society arises from reduced reward diversity that leaves more as the main source of positive reward-prediction error. The reviewer disputes this (point 3) and complains that I do not explain how to measure the effects of shrinking reward diversity. A nice example of such a measurement has just appeared: male fruit flies, deprived optogenetically of the reward from ejaculation, drink more ethanol (Zer-Krispil et al. 2018 Ejaculation induced by the activation of Crz neurons is rewarding to *Drosophila* males Current Biol. 28:1445-1452).

Regarding humans, it seems as obvious today, as it did to Adam Smith, that simplified tasks, which constitute many of the ‘jobs’ in modern society, do not nearly approach the challenges posed by foraging – for which our brain evolved. I have expanded slightly the text to explain this. In short, the chain of reasoning about the reward system and the consequence of shrinking reward diversity is well founded in current thinking.

The reviewer’s comment about a laborer in Cheops’ pyramid (point 3) is not relevant because I do not try to characterize every type of labor across 150 millennia. I simply compare foraging to modern life. However, to reduce a possible source for minor dispute, I have deleted my minor comment about “monumental constructions”. Regarding the reviewer’s comment regarding the Opium Wars (point 7): of course, the reward system can be manipulated for devious ends. Provide cheap opium and many will become addicted; the same is true today for cheap fructose syrup. But that is not relevant to my hypothesis, namely that the current cause of unbounded consumption is the loss of reward diversity.